



# Probabilistic forecasting of wind power production losses in cold climates: A case study

Jennie P. Söderman[1], Heiner Körnich[2], Esbjörn Olsson[2], Hans Bergström[1], and Anna Sjöblom[1]

[1]Uppsala University, Uppsala, SWE
[2]Swedish Meteorological and Hydrological Institute, Norrköping, SWE

*Correspondence to:* Jennie P. Söderman (jennie.perssonsoderman@geo.uu.se)

**Abstract.**

The problem of icing on wind turbines in cold climates is addressed using probabilistic forecasting to improve next-day forecasts of icing and related production losses. A case study of probabilistic forecasts was generated for a two-week period. Uncertainties in initial and boundary conditions are represented with an ensemble forecasting system, while uncertainties in
the spatial representation are included with a neighbourhood method. Using probabilistic forecasting instead of one single forecast was shown to improve the forecast skill of the ice-related production loss forecasts and hence the icing forecasts. The spread of the multiple forecasts can be used as an estimate of the forecast uncertainty and of the likelihood for icing and severe production losses. Best results, both in terms of forecast skill and forecasted uncertainty, were achieved using both the ensemble forecast and the neighbourhood method combined. This demonstrates that the application of probabilistic forecasting
for wind power in cold climate can be valuable when planning next-day energy production, in the usage of de-icing systems, and for site safety.

## 1 Introduction

Wind power production in cold climates experiences significant problems with production losses because of icing. Icing on
the turbine blades reduces the energy production due to change in the aerodynamic balance, generation of vibration and increased load (Kraj and Bibeau, 2010). Furthermore, site safety is an issue since falling ice poses a threat to the public and to maintenance. Despite these complications, a substantial part of the wind power production is located in cold climate regions. This geographical choice results from both the possible higher production in lower temperatures where the air is more dense than in warmer regions, and from the low population density, which reduces public safety risks and disturbance. According
to the World Market Update 2012 (Wallenius et al., 2013), more than 24 % of the global wind energy capacity was located in cold climate regions at the end of 2012 and most of these turbines experience between light to heavy icing. In order to plan for



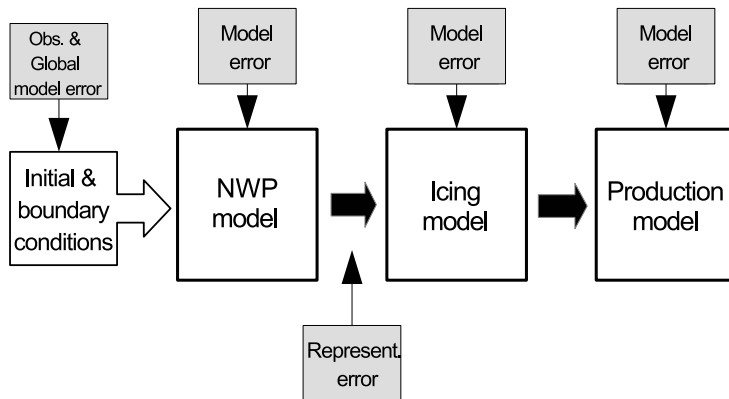

**Figure 1.** The modelling chain for forecasting icing related production losses. Uncertain parts are pointed out.

next-day energy production and site safety, short-range forecasts of icing and related production losses are vital tools for the energy market.

Forecasting icing and related production losses is challenging due to uncertainties in both the meteorological conditions and the modelling of the involved processes, e.g (Bergström et al., 2013; Davis et al., 2014). A common approach for the modelling
chain is shown in Figure 1. A Numerical Weather Prediction (NWP) model is used to forecast meteorological parameters that serve as input to an icing model. Finally, a statistical production model calculates the icing-related production losses from the forecasted wind and icing. It should be noted that all steps in the modelling chain of Figure 1 contain uncertainties, either in the model formulation or the required input data. The NWP model and the icing model suffer from lack of knowledge about the physical processes, from the numerical discretization as well as from simplifications to make the models computationally
affordable for operational usage (Yano et al., 2015). Initial conditions for the NWP model are also uncertain due to errors in the meteorological observations and assumptions in data assimilation methods (Megner et al., 2015). Forecasting wind power production requires high horizontal resolution in the order of kilometres to capture wind fields at 100 meters above ground in the Scandinavian mountains and also to model small-scale atmospheric phenomena leading to icing (Bergström et al., 2013). Due to the computational needs of such models, the domain size is limited and lateral boundary conditions are provided by a
host model, adding further uncertainties in the modelling chain. Finally, a statistical production model is based on a limited set of previous forecast validations and on assumptions about the functional relationship between wind, ice and production, also resulting in uncertainties. Because of these error sources in the modeling chain, the forecasted production losses are uncertain.





This issue has been addressed by using different NWP models that result in different estimations of ice load and icing intensity, and hence production losses (Ronsten et al., 2012; Bergström et al., 2013). Here, we address this problem in another way.

A common approach for uncertainty quantification in NWP is to use ensemble forecasting (Leith, 1974). It is known that in a non-linear dynamical system such as the weather, the largest uncertainty results from initial errors growing rapidly with

forecast time. These errors can be represented by re-running the model multiple times, starting from slightly different initial conditions (Leutbecher and Palmer, 2008). This collection of forecasts is generated by an Ensemble Prediction System (EPS). Global EPS have been run since the early 90s at e.g. the European Centre of Medium-range Weather Forecasts (ECMWF). In the beginning the focus of ensemble forecasting lay mostly on medium-range global forecasting. In the last 10-15 years, meso-scale EPS has been developed at different weather centers (e.g. the Metoffice (Bowler et al., 2008) and NOAA (Du et al.,

2003)) addressing the uncertainties on the short range, i.e. during the first 48 hours of the forecast.

An ensemble forecast can be utilized in several ways. The ensemble mean generally has a lower error than a single forecast, because the less predictable parts have been filtered out when averaging the ensemble members (WMO, 2012). This method was used by Al-Yahyai et al. (2011) to estimate the average wind speed over Oman in different seasons, and the ensemble mean reduced the forecast error compared to a single forecast. The difference between, or the spread of, the ensemble members can

represent the uncertainty of the forecast. An ensemble forecast can also be used probabilistically to estimate the likelihood of a specific event, for example the timing of a sudden change in wind speed (Schäfer, 2014). Furthermore, EPS for short-range forecasting has been investigated for wind energy purpose, e.g. (Pinson and Kariniotakis, 2010; Traiteur et al., 2013). Traiteur et al. (2013) studied the use of short-range EPS for 1-hour wind speed forecasting and showed that a statistically calibrated EPS outperforms other forecast methods. Here, we will employ an EPS for forecasting icing-related wind power

losses.

An additional uncertainty arises due to the fact that kilometre-scale phenomena, such as convective clouds, have faster forecast error growth than phenomena on larger scales, such as the position of a low-pressure system. Thus, a forecasted small-scale cloud can be misplaced by some tens of kilometres generating an error in spatial representation. This uncertainty has implications for how the forecast can be interpreted at a specific wind turbine location. Mittermaier (2014) suggested the

use of the neighbourhood method in order to address this misplacement of small-scale features in forecasts. In this method, a selected number of the surrounding grid points to an observation site are treated as equally likely forecasts. These forecasts can then be used in the same way as an ensemble, accounting for the uncertainties in the representativeness at each wind turbine location. An approach to generate probabilistic forecasts of wind power called adapted resampling was already used earlier by Pinson and Kariniotakis (2010), where the value of probabilistic forecasting for trading and wind power management was

demonstrated and it was suggested that methods of wind power forecasting should not rely directly on point forecasts as input.

In this case study, probabilistic next-day forecasts for wind power in cold climates has been run for a two-week period in winter 2011/12. The modeling chain (Figure 1) is extended with meso-scale ensemble forecasts and the neighbourhood method. These extensions address the uncertainties in the initial and boundary conditions as well as the representation error of the NWP part. In order to examine the impact of these terms separately, different combinations of ensemble forecasting and the

neighbourhood method are examined as the uncertainty quantification of the forecast for icing and related production losses.



Thus, it will be investigated whether these probabilistic methods add value to specific challenges of wind power forecasting in cold climates.

The models in the modelling chain are described in Section 2.1. The specific experiment period and available observational data are described in Section 2.2. The different approaches for the uncertainty quantification are presented in Section 2.3 and the verification methods in Section 2.4. The results in Section 3 are divided into two parts: Meteorological parameters in Section 3.1 and forecasts of icing and production losses in Section 3.2. Concluding remarks are given in Section 4.

## 2 Method

### 2.1 Description of the Models

#### 2.1.1 NWP model

As the NWP model, the Ensemble Prediction System HarmonEPS is used. HarmonEPS is a non-hydrostatic, convective permitting model intended for predictions of probabilities of high-impact weather events. It is based on the ALADIN-HIRLAM shared system and contains two packages of physical parameterizations, AROME and ALARO, of which the AROME package (cy38h1.2) was used here in the HARMONIE-AROME configuration (Bengtsson et al., 2017). HARMONIE-AROME has been used for operational weather forecasts at the Swedish Meteorological and Hydrological Institute (SMHI) since 2014 (Müller et al., 2017), also as an ensemble in HarmonEPS since 2016. In the present study, the horizontal resolution of the model is 2.5 km and it has 65 vertical levels. The model domain can be seen in Figure 2. The lateral boundary conditions come from the global EPS at ECMWF with an horizontal resolution of 30 km and are updated at 00 UTC and 12 UTC. A spinup period of three weeks were used to generate the start of the forecast period. The HarmonEPS ensemble consists of 10 perturbed members and one control member. The number of ensemble members were chosen based on a short-range EPS study, by Du et al. (1997), where it was shown that 8-10 ensemble members are sufficient for at least 90 % of the possible benefit of using an EPS. Since an EPS is computationally demanding to run, 10 members were therefore considered to be sufficient for the present study. The control member is using 3D-variational data assimilation of conventional observations as well as satellite observations from the instruments AMSU-A and AMSU-B, with 6 hour cycling, and the perturbations for each ensemble member come from the boundary conditions of different members of the ECMWF EPS that are added to the analysis of the control member. As the control member has no perturbations on the initial and boundary conditions, it should statistically outperform the other ensemble members.

#### 2.1.2 Postprocessing of the NWP data

Despite of increasingly higher resolution, the NWP models still lack some topographic details. The height of mountaintops in the model terrain remains in most cases below the actual height. The NWP output parameters are therefore adjusted to account for the difference between model terrain and real topography. Adiabatic lifting between model terrain and real terrain





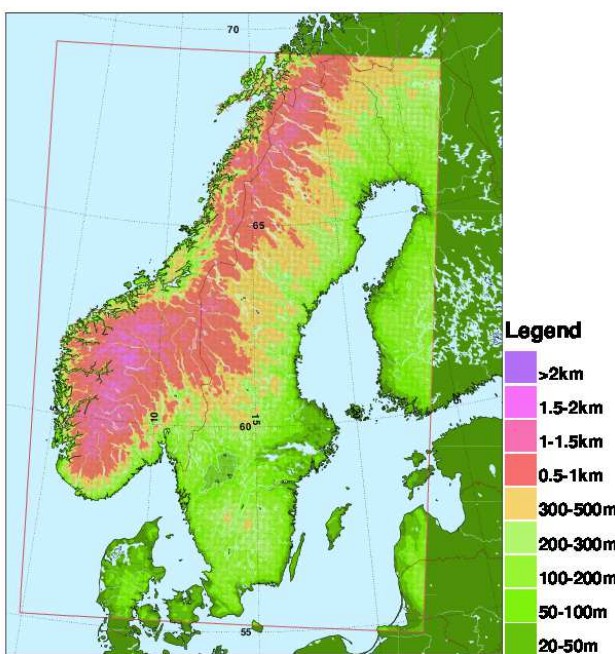

**Figure 2.** NWP model domain. Colours represent topography described in legend.

has yielded unsatisfying results. Thus, the following vertical interpolation is used for the NWP output; here for the example of temperature (T):

$$T_i = \frac{T(h_m + \Delta h + h_{nacelle}) + T(h_m + h_{nacelle})}{2} \tag{1}$$

where $T_i$ is the vertically interpolated temperature, $h_m$ is the model terrain height, $\Delta h$ is the difference between the real
5 terrain height and the model terrain height, and $h_{nacelle}$ is the height of turbine nacelle. Effectively, the forecast made at the actual terrain height plus nacelle height, and the forecast made at the model terrain height plus nacelle height are averaged in Eq. 1. Using only the forecast data at the actual height over sea level for the wind turbine nacelle can result in using atmospheric parameters well above the terrain compared to the turbine height. On the other hand, using only the forecast data at the height of the turbine nacelle above the model terrain can result in atmospheric data at lower height than the actual height of the wind
10 turbine. As important parameters for wind power, such as wind, temperature and moisture, vary strongly with the distance from the surface, a realistic choice for the vertical interpolation is vital. By choosing the averaging method from Eq. 1 for height interpolation these error sources were reduced (not shown here). This interpolation is done for all parameters. It should be




noted that in the case where model terrain height is higher than the real terrain height, only the model terrain height is used. However, no such grid point was found in the current study.

As a bias in the forecasted meteorological parameters could be damaging to the results, bias correction was considered at each time step before using the parameters as input to the icing and production model. However, no such correction is made,

since no reliable information on bias for each station at each time step could be derived.

The neighbourhood method following Mittermaier (2014) is used in order to capture the local uncertainty of the NWP data. Averaging forecasts made at several grid points around an observation site results in a better forecast than one single forecast from kilometre-scale NWP. Furthermore, the spread of the forecast from the neighbouring grid points provides an estimate of the forecast uncertainty. Here, the 25 nearest grid points to an observation site are chosen as equally likely forecasts. Since

these grid points are some kilometres apart from the turbine site, the height difference of the local topography can be several hundred meters. Two versions of the neighbourhood method was tested. In the first version, which is also the version presented in the result section (Section 3.2.3), the same height above sea level was used for all grid points, resulting occasionally in a height above ground much larger than the wind turbine height. The other version tested was a terrain-following method, where the same height above ground was used for all grid points. The different versions are further discussed in the result section.

### 2.1.3   Icing Model

The meteorological parameters forecasted by the NWP model are used to calculate ice loads utilizing a cylindrical ice accretion model, based on an equation often referred to as the Makkonen equation:

$$\frac{dM}{dt} = \alpha_1 \alpha_2 \alpha_3 W v A \tag{2}$$

where $M$ is the mass of ice, $t$ is the time, $\alpha_1, \alpha_2$ and $\alpha_3$ are efficiency coefficients, $W$ is liquid water content, $v$ is wind

speed and $A$ is the cross-sectional area of the cylinder on which the ice accumulation is calculated (Makkonen, 2000). The efficiency coefficients take into account aspects of the object where the ice is accumulated, such as the possibility for water adhering to the surface. A detailed description of the coefficients is found in Makkonen (2000). Meteorological inputs needed for the ice calculations are temperature, wind speed, liquid water content and median volume droplet size. The latter is not directly available from the present NWP models, so a value is estimated using the liquid water content and the concentration

of droplets. The concentration of cloud droplets is set to a constant of $100 \ cm^{-1}$ except in the case of precipitation, when the number of droplets is instead based on output from the NWP model.

In addition to the original Makkonen equation, which only accounts for ice accretion due to cloud water, ice accretion due to cloud ice, snow and rainwater are included in the icing model. It is assumed that snow and graupel are only contributing to the ice accretion if rainwater is also present since dry snow easily rebounces after the collision with the turbine. The sticking

efficiency $\alpha_2$ is different for snow and graupel compared to cloud water. Based on Nygaard et al. (2013) where both $\alpha_2 = 1/v^{0.5}$ and $\alpha_2 = 1/v$ were discussed, $\alpha_2 = 1/v^{0.75}$ is used here. The different forms of water in the cloud are fed separately into the equations using their forecasted concentrations from the NWP model. The equations for calculating droplet number





concentrations for cloud ice, rain, snow and graupel have been taken from the AROME microphysics scheme (Seity et al., 2011). The median volume droplet diameter is calculated according to a scheme for cloud water by Thompson et al. (2008).

Formulas for melting, shedding, sublimation and wind erosion are also additions to the model compared to the original Makkonen equation. Melting of ice is calculated using an energy balance equation, which includes an empirical ice shedding.

The sublimation is calculated using eq. 19 in Mazin et al. (2001) which utilizes wind speed and relative humidity in the calculations. The wind erosion is calculated by multiplying a hourly rate coefficent of $10\ gm^{-2}(ms^{-1})^{-1}$ with the wind speed when the wind speed is greater than $5\ ms^{-1}$, otherwise the erosion is zero. Here, only the wind speed at the nacelle is used. If the actual winds at the rotating wind turbine blade is used, the wind erosion coefficient needs to be reduced approximately by a factor of ten (Davis et al., 2016).

A more detailed documentation of the icing model can be found in Bergström et al. (2013). It should be noted that there are some differences in the model version since used in Bergström et al. (2013) as described above, i.e, the additional wind erosion calculations and the height interpolation.

### 2.1.4 Production Model

The production model consists of two parts, one part for the potential production, and one for the production loss.

Ice-free seasonally varying effect curves were calculated for each wind turbine at every wind park, using a minimum of two years of production and wind speed observations. Only production observations with temperatures above $5°C$ were used to ensure that the blades are free of ice. The effect curves are then used with forecasted wind speed to calculate the potential production.

The production loss forecast requires the modelled parameters of ice intensity, ice load and wind speed as input. It uses two
matrices separating the losses due to ice load and icing intensity (Bergström et al., 2013). For a specific wind speed, ice load and icing intensity, the model yields a production loss in percent. Only one of the matrices is used for each forecast depending on which gives the highest production loss. The matrices were constructed manually using hindcasts of ice load and wind speed from a two-month period in 2010 combined with observed production values from one specific wind park which showed good agreement between observed and modelled icing. Due to contractual reasons the wind park must not be specified. The
matrices were generated by fitting 0 and 100 % losses against observations and then by interpolating the values inbetween. The empirical functions for production loss, determined for one specific wind farm, were used in the production forecasts for all wind farms. Generally, the icing intensity influences the production losses more than the ice load (Bergström et al., 2013). Finally, the potential production is combined with the forecasted loss to provide the actual forecasted production output.

The observed production loss was calculated from the ratio between the observed production and the potential production,
given the observed wind speed and the ice-free effect curves. A value of 30 % means 30 % less production than the potential production.





## 2.2 Experiment period and available data

The case study is based on a two-week period. HarmonEPS was run 26 December 2011 to 7 January 2012, initializing a 42-hour long forecast at 00, 06, 12 and 18 UTC, with hourly output. The forecasted values from 18 to 42 hours starting from 06 UTC are used as the "next-day" forecast in the evaluation of the forecasted production losses and production.

Observations are available from 10 wind parks in Sweden that must not be specified due to contractual reasons. At some sites the observations were made from meteorological masts and at others at the turbine nacelle. All 10 sites measure temperature, wind speed, wind direction, relative humidity, pressure and ice load. The sites are located from northern to southern Sweden between 250 and 1,000 m above sea level and the measurement height above ground is between 60 and 150 m. From four of the sites, production data from each turbine is also available for the period.

The meteorological parameters are measured every 10 minutes with the multi-instrument Quatro-Ind H (Lambrecht Meteorological Instruments, Germany), except at one site where the multi-instrument WXT510 (Vaisala, Finland) is used. Since the NWP forecast output is hourly, only the 10-minute observations at every full hour are used in the verification.

The ice load is measured with an IceMonitor Combitech AB (2016). The IceMonitor measures ice on a rotating cylinder according to ISO 123494 specifications ISO 12494 (2000). Some problems have been identified with this instrument. One is

that it may stop rotating, and the ice is then only accumulated on one side of the cylinder. Another issue is that the ice can cause the rod to lift and thus to measure an incorrect ice load (Thorsson et al., 2015). Therefore, it is difficult to use the ice measurements quantitatively, but they are still used here to get an approximate observation of the icing.

The following quality controls were conducted for the observations of temperature (T), wind speed (WS) and relative humidity (RH). Values for $T < -40\,°C$, $WS < 0\,m\,s^{-1}$ and $RH < 0\,\%$ are removed from the data set as being unrealistic. Also,

if the standard deviation of the 10-minute averaged wind speed was zero, the observation was removed. Finally, in order to remove unrealistic jumps in the observations: if the difference between the current observed value and the next deviates more than three times the standard deviation of this difference for the entire period, the observation was removed. For one site, we found that wind speed measurements were unreliable, as the instrument was heavily affected by icing. This influence could be confirmed in webcam pictures. Thus, this station including its production data was omitted.

For the production data, data was only used when no error code from the site was given. Thus, the reduction in observed production should be caused by icing.

## 2.3 Uncertainty quantification approaches

Two methods for uncertainty quantification are employed in this study; ensemble forecasting and the neighbourhood method.

Ensemble forecasting is used here to account for uncertainties in the initial and boundary conditions within the NWP model

HarmonEPS as described above. The possibility to include model errors in the ensemble prediction system was omitted in order to stay as close as possible to the operational setup of the NWP model. The neighbourhood method accounts for uncertainties in the representativeness of the forecast for a specific location of the wind turbines, e.g. the site of a wind turbine.





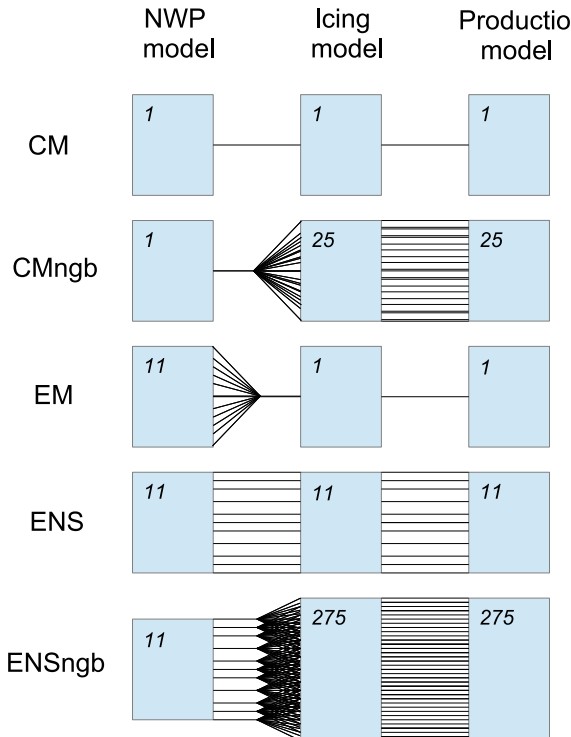

**Figure 3.** Uncertainty quantification approaches. The commonly used approach CM and the four uncertainty quantification approaches. CM: Control member, a single forecast used in each forecast step. CMngb: Control member together with the neighbourhood method. EM: Ensemble mean, averaging the ensemble members after the first modelling step. ENS: Ensemble, using all ensemble members throughout all modelling steps. ENSngb: Ensemble together with the neighbourhood method. The number in each box represents the number of forecast members in each forecast step.

The control member (CM) from the ensemble forecast is used as baseline, as it reflects the use of a single deterministic forecast. In order to quantify the role of the different uncertainty sources on the forecast uncertainty, the benefit of using different combinations of the two above methods compared to baseline is investigated.

Four different combinations of the two methods were studied. These four uncertainty quantification approaches are in addi-
tion to CM presented in Figure 3 and are described below.

  – CMngb (Control Member Neighbourhood): The model chain starts with a single NWP forecast, namely the control member from the ensemble forecast. Next, the neighbourhood method is added providing multiple forecast input to the icing and production model. This approach results in 25 forecasts from the 25 neighbouring grid points.





- EM (Ensemble Mean): The model chain starts with the ensemble forecast from HarmonEPS. The 11 ensemble members (the 10 perturbed members and the control member) are then averaged before the icing forecast, providing the single statistically best meteorological input for the icing and production model. The uncertainty of the icing and production forecasts cannot be determined.

- ENS (Ensemble): The first modelling step is based on the ensemble forecast. All ensemble members are then used each as input to the icing and production loss models, resulting in 11 forecasts of icing and production loss. The 11 forecasts give an estimation of the uncertainty in the icing and production forecasts.

- ENSngb (Ensemble Neighbourhood): The ensemble and the neighbourhood method combined. The neighbourhood method is added to each ensemble member after the NWP model step, resulting in 25 forecasts for each ensemble
member and a total of 275 forecasts used as input to the icing and production model.

## 2.4   Verification methods

The forecast skill is assessed by the root mean squared error (*RMSE*), the mean forecast error called *bias*, and the unbiased forecast error, ie. the standard deviation (*std*) of the forecast error. They are connected by:

$$RMSE^2 = bias^2 + std^2 \qquad (3)$$

The calculation of the forecast error as the deviation between forecasted and true value uses an observation as replacement for the truth. Thus, the observational error needs to be taken into account in the calculation of the error terms. Due to the lack of a consistent estimate for the observational error, the observational error is neglected. This leads to an overestimation of the RMSE, bias and std.

The magnitude of the bias of the meteorological parameters in this study varied between the observation sites and with
forecast lengths. Thus, the average bias for each forecast length was estimated and removed from the $RMSE$ to get $std$.

By averaging the ensemble members for the same valid time, they can be treated in the same way as a single forecast, and the same skill scores can be used allowing to compare the skill of the different probabilistic approaches with the skill of the single deterministic forecast denoted as CM.

The spread of the ensemble forecast contains important additional information compared to a deterministic forecast, such as
a situation-dependent estimate of the forecast uncertainty. The spread $SPRE$ is defined as:

$$SPRE = \sqrt{\frac{1}{M-1}\sum_{i=1}^{M}(x_i - \bar{x})^2} \qquad (4)$$

where $x_i$ is the i-th forecast member, $\bar{x}$ is the ensemble mean and $M$ is the number of forecast members.





In a perfect probabilistic forecast, any of the simultaneous forecast members would be statistically indistinguishable from the truth. If this is not the case, the forecast uncertainty is over- or underestimated, i.e. providing too wide or too narrow range of forecast outcomes, respectively. In order to verify the ensemble spread ($SPRE$), it is compared to the forecast skill of the ensemble mean in terms of unbiased forecast error (*std*) following Johansson (2017).

Both the $std$ and the $SPRE$ are statistically expected to increase with increasing forecast length. In a perfectly calibrated forecast, the spread should be as large as the skill. The so-called spread/skill-relationship of the forecasts, $SPRE/std$, should therefore equal one. As the unbiased forecast error $std$ is overestimated due to the neglected observational error as mentioned above, the spread/skill-relationship will consequentially be underestimated. This error could be corrected with an appropriate estimate of the observational error (Schwartz et al., 2014).

The skill of the forecasts made with the different approaches from section 2.3 and the spread/skill relationship are presented below. Since a relatively short period of forecasts is studied with only two icing events, it is not possible to test the significance of the statistical measures. The results should therefore only be considered to show the potential benefit of the use of probabilistic forecasting for wind power in cold climates.

## 3 Results

### 3.1 Meteorological parameters

For the two-week period examined in this study, the skill of the basic meteorological model performance is presented in Figure 4 in terms of bias and unbiased forecast error for relative humidity, wind speed and temperature for the 42-hour forecasts. The statistics is based on all forecasts at all 10 observation sites for each forecast length. The unbiased forecast error averaged over the forecast window amounts to about 7 %, 2.3 $m$ $s^{-1}$ and 0.9 $°C$ for relative humidity, wind speed and temperature,

respectively. The forecast error is expected to increase with forecast length, which is the case for temperature (Figure 4c). The unbiased forecast error of the wind speed and relative humidity (Figure 4a/b), however, displays a flat shape suggesting that the error is saturated during the forecast window. This behaviour points to a difficulty in the analysis, i.e. the initialization of the wind field in the model. One aspect in this regard is the lack of wind observations in the planetary boundary layer to initialize the meso-scale wind field.

The temperature bias increases from $-0.3$ $°C$ to close to $-1$ $°C$ after 42 hours (Figure 4c). This behavior might be caused by a spin-up problem in the model, since it is changing with forecast length, and thus might not be related to the warm turbine affecting the measurements, which can generally cause a negative temperature bias. The bias of the wind speed (Figure 4b) decreases slightly with forecast length from $1$ $m$ $s^{-1}$ to $0.5$ $m$ $s^{-1}$ after 42 hours. The bias in the relative humidity (Figure 4a) displays only small changes with forecast length and remains generally small with absolute values less than 1 %.

In general, the meteorological model displays good performance for the basic meteorological parameters. The lack of forecast error growth in wind speed and relative humidity suggests that the model forecast could be improved by assimilating more observations, especially for wind and humidity. A better initial state of the forecast might even improve the bias behavior.





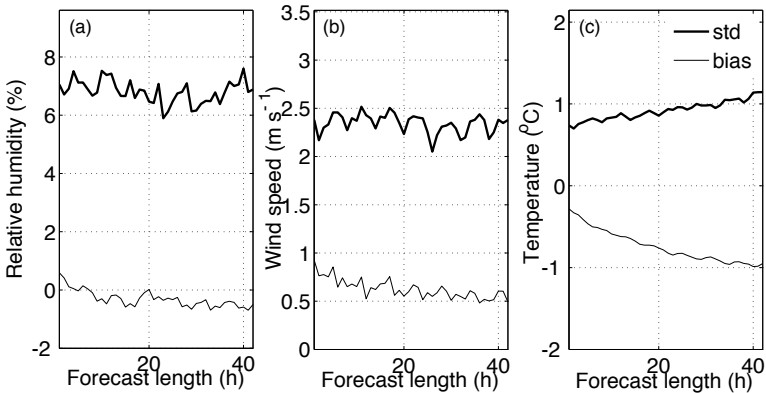

**Figure 4.** CM: Unbiased forecast error $std$ and bias for a) relative humidity in percent, b) wind speed in m s$^{-1}$ and c) temperature in $^{o}$C for increasing forecast length. The statistical measures are calculated for the two-week period and averaged over all available stations. The meteorological observations at 60 to 150 m height above ground from 10 sites are used.

The meteorological performance of the different uncertainty quantification approaches in terms of spread and skill is shown in Figure 5. The skill as defined as the unbiased forecast error $std$ and spread for the temperature forecasts is displayed for each approach. The baseline approach of a deterministic forecast (CM) shows the largest forecast error. For the examined period, the benefit of using neighbourhood and/or ensemble methods can already be seen in the first hours of the forecasts,

and increases with forecast length as expected. The CMngb has the smallest improvement of forecast error, but still suggesting that the neighbourhood method is valuable, if an ensemble forecast is not available. Even higher improvement is achieved by the approaches ENS and EM that provide by construction the same values here. The largest reduction of the forecast error is achieved using both the ensemble forecast method and the neighbourhood method (ENSngb), with an average reduction of 9 % for the temperature forecast error averaged for all sites and all forecasts. Similar behavior was found for the forecasted wind

speed and relative humidity (not shown).

Figure 5 also displays the forecast spread of the approaches ENS, ENSngb and CMngb. The spread is always clearly lower than the unbiased forecast error, which means that the forecast uncertainty is underestimated in all approaches. This behaviour might result from neglected uncertainty sources in the modelling chain (Fig. 1) The employed ensemble represents uncertainties in the initial and boundary conditions, but it does not take into account uncertainties in the model physics or numerical

formulations. There are methods to account for these uncertainties, such as stochastic physics, which increase the spread (Bouttier et al., 2015), but they are not in the scope of this study. It is also important to consider the observational error when validating the spread-skill relationship of the forecast, as discussed in Section 2.4. Since the forecast error estimate is not corrected for the observational error, the underdispersiveness of the approaches is not as large as it appears in Fig. 5.

On its own, the ensemble forecasting method (ENS) has better skill, spread and spread/skill relationship than the neigh-

bourhood method (CMngb) for this period, around $0.6$ and $0.3$ respectively for all meteorological parameters. Specifically, the spread/skill relationship improves from around $0.3$ for CMngb to around $0.6$ for ENS for temperature, wind speed, and relative



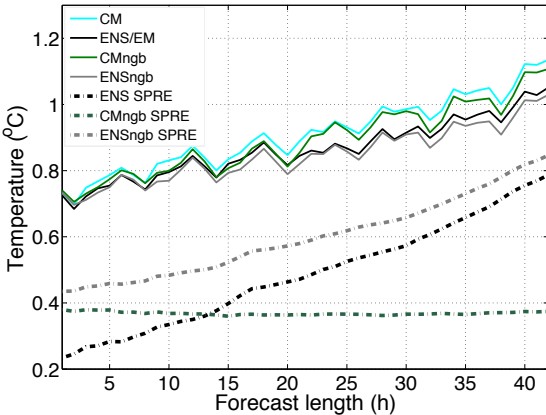

**Figure 5.** Unbiased forecast error $std$ for all uncertainty quantification approaches for the temperature in $^oC$ and the spread, $SPRE$, for the three approaches consisting of multiple forecasts as a function of increasing forecast length. The $std$ is calculated for the average forecast in case of multiple forecasts i.e. approach CMngb, ENS and ENSngb.

humidity. The approach ENSngb results in the best spread/skill relationship, around $0.7$. This implies that the neighbourhood method is adding additional information about the forecast uncertainty, which is valuable for both the forecast skill and the forecast spread. The spread resulting from the neighbourhood method is constant with forecast length (CMngb in Figure 5), since it represents the internal variations of the weather over the scale of the neighbourhood domain, but it does not take into

account uncertainties due to initial boundary conditions, or model formulations that increase with increasing forecast length.

### 3.2 Forecasted Icing and Production losses

The forecasted icing is, as mentioned before, more difficult to validate since the observations of icing are unreliable. Alternatively, the forecasted production loss is also considered as a measure of the forecast skill of the icing. Results from two of the three sites, here called A and B, with consistent production observations will be presented in more detail and the average skill

and spread for the different approaches at the three sites are discussed.

#### 3.2.1 Site A

Two icing events were forecasted at site A during the two weeks. During the first icing event, around the 30/12, only a small amount of ice is forecasted, while during the second icing event, starting around 1 January, both the modeled and observed ice load amount to several kg m$^{-2}$. In Figure 6a the forecasted icing using the ENS approach and the CM approach is presented

together with the ensemble mean of the ENS approach and observed ice load. The spread of the ensemble members from the ENS, which displays the forecast uncertainty of the icing, are presented in Figure 6b. During the icing events, the spread





signals uncertainty in the ice amount. Largest forecast uncertainty is found at the end of the second icing event with around 1.5 kg m$^{-2}$. The magnitude of the overall forecast uncertainty, or spread, amounts to about 50 % of the ice load here.

In this period, first the build up and then the loss of ice happen almost simultaneously for all the ensemble members. This is especially visible in the spread for the second icing event with values close to zero at the start and end of the event. Forecasted

and observed production loss for the site allow studying the timing during the event in more detail (Fig.  7a). The forecasted production loss starts about 12 hours later than the observed production loss, pointing to a problem with the agreement between the ensemble members. A closer examination reveals low forecasted liquid water content in the beginning of the icing period by all the ensemble members (Figure 6c), resulting in no accumulation of ice by the icing model. Instead the ice starts to accumulate in the model when snow and graupel is forecasted, in addition to the cloud water. Interestingly all ensemble

members behave very similarly with this timing. This could be related to an error in the modelled cloud characteristics or in the icing model causing a too slow ice buildup.

Additionally, the ensemble fails to describe the end of the icing period (6-9 January) where the modelled production loss drops to zero for the remaining period, while the observed production loss is high around 8 Januray (Fig. 7a). The end of the period also lacks forecasted liquid water content, and thus no further buildup by the icing model is possible. Again, the

ensemble is overconfident with all members displaying a similar behaviour. By including the neighbourhood method to the ensemble, there is some improvement adding some forecasted liquid water content at the end of the period (Fig. 8). However, the amount of forecasted water content was still too low to generate any notable buildup by the icing model. In Davis et al. (2014) it was shown using a similar model that the resulting ice load was highly sensitive to a variation of the ingoing median volume droplet size of the water droplets, suggesting that this is an uncertain part of the icing model. Here, the droplet size is

calculated from the liquid water content before being used in the icing model.

A closer look at the weather situation shows that site A is affected by a frontal passage during the second week. The ensemble members all have the front passage at the same time, resulting in similar cloud cover and similar liquid water content. This problem results from the insufficient representation of the uncertainties in the boundaries. The boundary data from different members of the ECMWF EPS prescribes very similar surface pressure patterns for this event resulting in an overconfidence in

the arrival time of the cloud front. Furthermore, small-scales are initialized for all ensemble members in the same way using the control member analysis.

### 3.2.2   Site B

At site B only one icing event was forecasted and observed, starting already on the first days of the period. This site was not as strongly affected by frontal passages as site A, and had more general cloudiness during the period. Figure 9a-c shows the

forecasted and observed ice load, spread of forecasted ice load and liquid water content. In Figure 10a-b the forecasted and observed production loss as well as the related forecast spread is presented. The observed ice growth is first small, but with increasing pace after January 1, 2012, reaching the largest load of about 2.5 kg m$^{-2}$ on January 4 (Fig. 9a). The observed production loss shows a similar behaviour (Fig. 10). The forecasted ice and production loss displays a reasonable agreement. However, there are two interesting deviations. Firstly, in the first half of the time series the forecast is generally underestimating

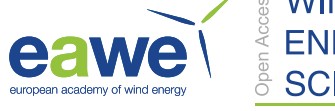

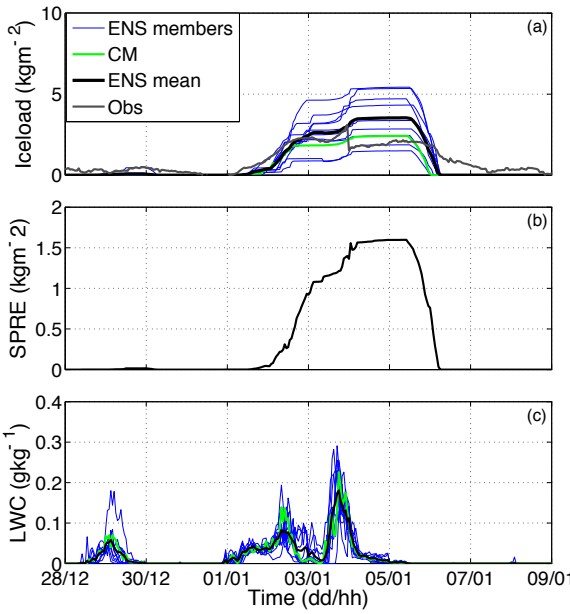

**Figure 6.** Site A. a) Forecasted ice load in kg m$^{-2}$. b) The related spread of the ensemble members in ENS. c) Forecasted liquid water content in g kg$^{-1}$ for the period. Blue lines are ensemble members, green line is the CM and black line is the ensemble mean of the ENS approach. The ice load observations are in grey in panel (a).

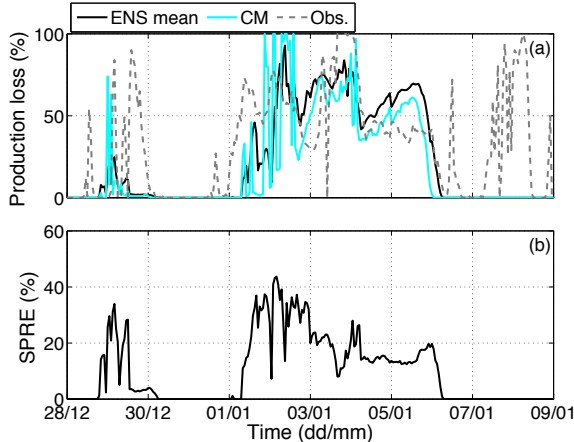

**Figure 7.** Site A. a) Forecasted production loss in percent. Black line is the ensemble mean using ENS, green line is the CM and dashed grey line is the observed production loss. b) The related spread of the ensemble members in ENS.

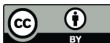


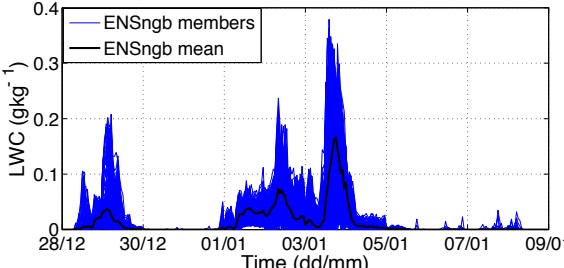

**Figure 8.** Forecasted liquid water content in g kg$^{-1}$ using the ENSngb approach.

the observed production loss (Fig. 10), while the forecasted ice load agrees well with the observations (Fig. 9a). The reason of the disagreement seems to be related to deficiencies in our production model. Secondly, the observed drop in both ice load and production loss at the end of the period is delayed in all forecasts, opposite to the behaviour at site A.

The high forecast spread, or forecasted uncertainty of the icing forecast (Figure 9b) shows the value of having a probabilistic

forecast. This variation of the ensemble members is probably due to a large variation of the liquid water content (9c). The benefit of an ensemble in this case can also be seen in Figure 10a where the ensemble mean (black line) of the production loss forecast starts to decrease, getting closer to observations, while the CM is remaining nearly at 100 %. The forecast spread of the production loss is also increasing during the last day of the period, suggesting an enlargened uncertainty in the forecast. Generally, forecasting the end of an icing period has been shown to be challenging due to the difficulties in modelling ice loss

(Davis, 2014). On the other hand the start of the stronger icing period from the 3rd of January is well timed in the forecast data, and furthermore, the spread of the production loss forecast increases from approximately 5 % to 30 % simultaneously. This increased spread is a useful indicator for the uncertainty of the start of the ice period and provides additional information to the actual forecast.

### 3.2.3 Forecast performance of the different approaches

In Figure 11a the forecasted next-day production using the four different uncertainty quantification approaches are compared with the single forecast of CM and observed production for site A. For the approaches that generate multiple forecasts, the average of the forecast members is presented. The production is calculated for a 2 MW-turbine.

The different approaches are following the observed production most of the time. As the figure suggests visually and the RMSE calculation below confirms, the ENSngb mean is the most skillful approach for this site. A closer inspection of Fig. 11a

reveals some typically behaviour for the approaches. All approaches overestimate the production during the start of the second icing period around the 1 January, which agrees with the underestimated production loss in Figure 10a. The different model forecasts tend to overestimate the production occasionally. This probably happens due to an overestimation of the potential production, since the wind speed has a positive bias (figure 4b). A bias correction of the wind speed would be useful to reduce this error.





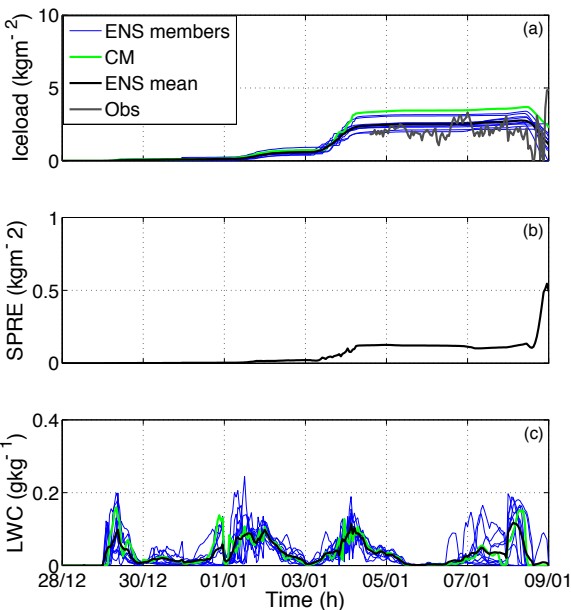

**Figure 9.** Same as Fig. 6, but for Site B.

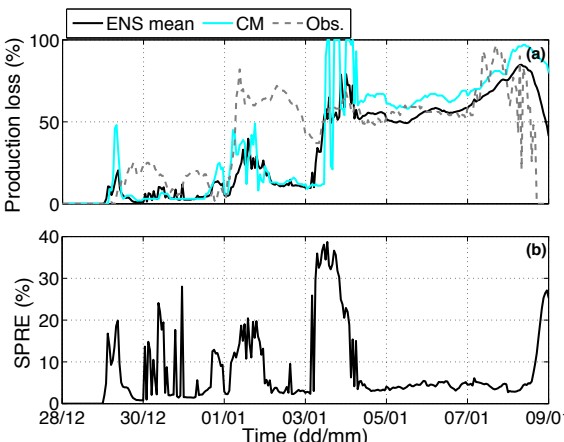

**Figure 10.** Same as Fig. 7, but for Site B.

In Figure 11a the single forecast of CM has stronger variations than the other approaches. These variations are partly a sign of uncertainty in the forecast and belong to unpredictable phenomena. They are filtered out when averaging over the members of the probabilistic approaches. However, it is important to realize that this filtering creates a so-called unoccupied average, ie. the smoothed state is unrealistic, since part of the variance is now included in the spread of the forecast members



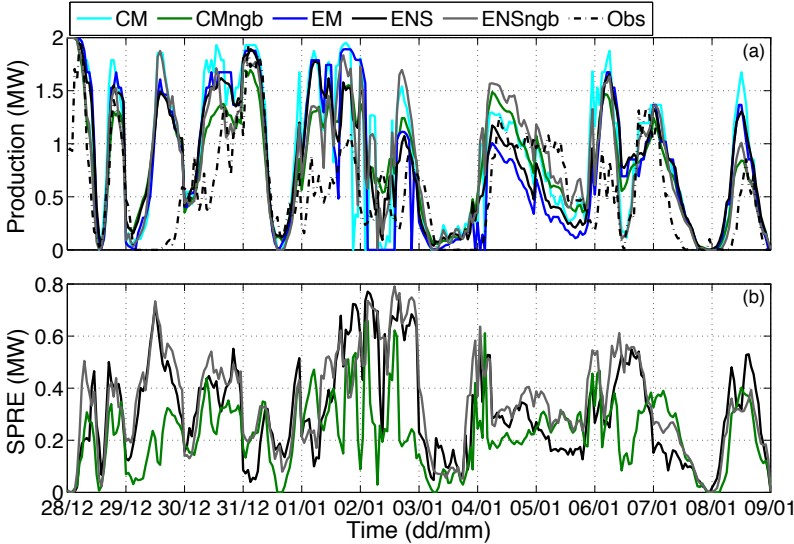

**Figure 11.** a) Production forecast in $MW$ at site A for the different approaches (Figure 3). For CMngb, ENSngb and ENS the lines present the average of the multiple forecasts. Observed production is given with the dash-dotted line. The production is calculated for a 2 MW-turbine. b) The spread of the multiple members in CMngb, ENSngb and ENS.

describing the forecast uncertainty. This spread for the uncertainty quantification approaches can be seen in Figure 11b. The ensemble (ENS) provides larger spread than the neighbourhood method applied to a single forecast (CMngb). Similar to the meteorological parameters, the largest spread, and thus forecasted uncertainty, is generally seen for the combination of ensemble and neighbourhood method (ENSngb). The average spread/skill ratio of the ENSngb production forecast amounts to

5 around 0.7, compared to 0.5 for ENS. The increased spread for the ENSngb should therefore provide the best estimate for the actual forecast uncertainty, even if the model is still overconfident.

As a summary of the forecast performance, Table 1 yields the mean error as $RMSE$ of the 06 UTC +18-42 h forecasts for the production loss and for the production, averaged over the three observation sites where production data was available. The forecast quality of the production forecast using the different approaches generally follows the order in the first step of the

10 modelling chain, i.e. for the meteorological parameters (Figure 3). The benefit of the neighbourhood method for the production loss forecast can be seen comparing the $RMSE$ forCM and the CMngb. However, the production loss $RMSE$ is the same for ENS and ENSngb, suggesting that the neighbourhood method neither contribute to or reduce the forecast skill when added to the ENS approach. Using the ENS or ENSngb approach compared to CM results in a reduction of the production loss forecast error from 26 to 21 %. It should be noted that the usage of the ensemble mean as the input to the icing and production loss

model (approach EM) deteriorates the forecast quality to 29 % compared to the ensemble-based approach ENS with 21 %, where the output from each member is calculated through the entire chain. This results from the non-linearity of the icing and





**Table 1.** Mean $RMSE$ of the different approaches for production and for production loss forecasts. The production is calculated for a 2 MW-turbine.

| Approach | Prod (MW) | Prod loss (%) |
|---|---|---|
| CM | 0.49 | 26 |
| CMngb | 0.44 | 23 |
| EM | 0.47 | 29 |
| ENS | 0.44 | 21 |
| ENSngb | 0.41 | 21 |

production loss model. The increased forecast quality for the forecasted meteorological parameters by the ensemble mean is lost by the usage of the unoccupied average as input into the icing and production model.

For the production forecast, the best forecast is provided by the ENSngb approach, while the worst comes from the single forecast of CM. Here, adding the neighbourhood method to the ENS approach reduces the $RMSE$ (Table 1). Using the
ENSngb approach compared to CM results in a reduction of the production forecast error from 0.49 to 0.41 $MW$ or by 16 % relative to the CM forecast.

The two different versions of neighbourhood methods, ie. terrain-following or constant height described in Section 2.1.1 displays some differences for the production forecast. Assuming that wind power is installed at higher elevation than the surroundings, the terrain-following version provides neighbours from lower absolute heights with higher moisture content, and
thus more atmospheric icing, and more surface-affected wind fields compared to the constant-height version. The quality of the resulting production loss forecast is very similar for the two versions, resulting in the same $RMSE$. For the final production forecast however, an improvement is seen from the constant-height version compared to the terrain-following one which can be traced back to better wind forecasts from the constant-height neighbourhood method.

It should be pointed out once more that the statistical significance of the results could not be assessed since this is a case
study with a limited sample size, but the results are consistent in the different analyses and supports theoretical expectations. The improved forecast skill of the production loss and of the production, using the four probabilistic forecasting approaches instead of the single forecast of CM, also suggests that the icing forecast is improved, even though it is not possible to validate with the available ice load observations.

## 4 Concluding Remarks

The problem of predicting next-day production losses due to icing of wind turbines has been addressed with the usage of probabilistic forecasting. Two methods, ensemble forecasting and the neighbourhood method, have been used in four differ-



ent uncertainty quantification approaches to produce probabilistic forecasts. Improved skill and estimations of the forecast uncertainty were both investigated in this two-week case-study. The main results are:

– Using probabilistic forecasting improves the forecast skill for the meteorological parameters, the icing and the icing-related production loss compared to the commonly used approach with one single deterministic forecast.

– The spread of the multiple forecasts can be used as an estimation of the forecast uncertainty, also for icing and related production losses. However with the current model setup, the uncertainty is underestimated both for the meteorological parameters and for the production.

– The approach where both the uncertainties in initial/boundary conditions and the representativeness of the wind turbine are represented, ENSngb, has the highest skill for the next-day production forecast. This suggests that both errors should
be taken into account when generating a probabilistic forecast.

Improving the skill by the use of an ensemble forecast is a useful contribution to wind power forecasting in cold climate. Additionally, a reliably forecasted uncertainty can be of great value for end-users as probabilistic forecasts of icing events and related production losses. Even though the spread of the forecast is too low and, hence, the forecast uncertainty underestimated, it could be utilized if the spread was calibrated, see e.g. Veenhuis (2012) or Sloughter et al. (2012). Knowing the likelihood for
icing, the end-user can employ site-specific cost-loss ratios in the decision making for the trading process, the use of de-icing systems and for the safety of people working at the wind farm.

To further develop the use of probabilistic forecasting in this area, it is important to note that we are not taking into account all of the uncertainties in the modelling chain, e.g. errors resulting from approximations made in the different models. It is known that the icing model contains numerous uncertain parameters, such as the sticking efficiency in case of snow and
wind erosion (Davis et al., 2014; Nygaard et al., 2013). The inclusion of these uncertainties into the entire modelling chain is currently ongoing research.

The weather is the most fundamental part of the modelling chain and due to its chaotic behaviour the focus here is the meteorological model. As the forecast skill for wind speed and relative humidity at nacelle height appeared to be saturated during the first 42 forecast hours, more effort is required in order to improve the initial state of the forecast, e.g. through
data assimilation methods or inclusion of local observations of humidity and wind, by radar and other instruments. From the meteorological spread/skill relationship, it can be concluded that also more spread is needed. The lack of spread in the ensemble was especially visible during a frontal passage where all members had very similar boundary conditions and thus the same timing of the frontal passage. Better spread from the boundary conditions can be achieved by a smart selection of the global host model ensemble members (Molteni et al., 2001). Furthermore, full-scale data assimilation for all ensemble
members, and not only for the control member, would allow for a better spread in the initial state of the ensemble members.

Finally, a NWP-ensemble forecast is computationally expensive to run and requires extensive infrastructure. Thus, often only national meteorological services can operationally produce ensemble forecasts. The open-data policy following the European INSPIRE directive will make these kinds of forecasts openly available. Then many users to include this forecast data into

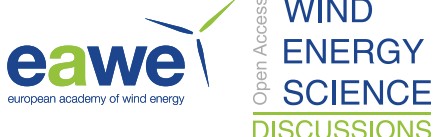

their own icing and production models which are far less computationally expensive. This will allow for a wide application of probabilistic forecasting for wind power in cold climates.

*Competing interests.*   No competing interests are present.

*Acknowledgements.*   The authors would like to thank the Swedish Energy Agency for financing the project within the program "Wind Power
5   in Cold Climate", project number 37279-1.





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
