# Peer review of "Probabilistic forecasting of wind power production losses in cold climates: A case study"

_Wind Energy Science, 2017_

## Referee Comment (RC1) · Anonymous Referee #1 · 20 Oct 2017

General comments: This article looked at applying two well-known probabilistic techniques from the atmospheric science community to the problem of production loss of wind turbines due to icing. The methods were applied on the NWP model simulations that were used as input to the icing model, which in turn is used as input to the power production model. The paper shows that a traditional model ensemble does not improve results when tested over two wind farms for a two week period, but that using forecasts from neighboring points does improve the forecast.

The article is well structured and the topic is of significant interest to the Wind Energy community. However, the methods are not described in enough detail for reproduction, and then observational data that is used is not described sufficiently nor of sufficient length for it to provide a good validation of the methods being tested. Finally, the paper

mentions that the cloud parameters LWC and MVD are key inputs to the icing model, but they are not analyzed in terms of the spread of the probabilistic approaches, which seems like an important metric to analyze.

Specific comments:

1. Section 2.1.1: How does the HarmonEPS used in this study differ from the operational usage of SMHI?

2. P4 L23-24: Do I read correctly that the models are all initialized using the ECMWF control member analysis, and only differ due to boundary conditions?

3. P5 L12: It would be interesting to see the impact of this averaging on the results. You state they are not here, but perhaps you could at least mention the improvement in error for the different variables in the text.

4. P5 L12: What are all parameters, and which parameters was the lifting validated against?

5. P6 L31-32: You note that different forms of water are fed separately through the model, how exactly is this carried out? Are you just running the model 4 times and summing the results? It is unclear to me how that would impact the accretion efficiency, which is a heat balance that depends on the mass flux of water impacting the structure.
6. P7 L2: Is the MVD only calculated for cloud water?

7. P7 L4: What is the empirical ice shedding model?

8. P7 L15: In not familiar with the term effect curve, is this just the power curve?

9. P11 L26: You mention that there could be an issue with warm turbines, but you mentioned that you have a mixture of mast and nacelle data. Did you investigate if the bias was different between the two sources?

10. P12 L10: could you list the % improvement of the wind speed and RH?

11. P13 L9: The shift from 10 sites to 3 was confusing when reading this paper, perhaps remind the reader that you only had production data at 4 sites, and had to discard one. I am not sure why you only looked at two of the three sites here though.

12. Section 3.2: You discuss quite often the LWC values and how they impact the ice growth. It is unclear to me what this parameter is, since your model includes two types of liquid water hydrometeors and two solid hydrometeors. Could you describe what it is, and if it doesn't already make sure it includes all of the relevant hydrometeors.

13. Section 3.2: In the experiment period and available data section, you mention that you have production data from each turbine at each site. It is unclear how you aggregated the production data to get a singular value. Can you state approximately how many turbines were at each of the two sites you used?

14. While it is understandable that you cannot list the wind farms themselves, can you at least describe how far apart they are, and if they are exposed to similar weather patterns.

15. Fig 6. It is hard to see the observations clearly, perhaps you could use a color like red that would stand out more.

16. Figure 9: Why was there no ice during the beginning of the period, even though there was still a fairly large amount of LWC?

17. Table 1: What is the mean RMSE the mean of?

---

## Author Comment (AC1) · 24 Oct 2017

Thanks to referee#1 for the comments on our paper!

We will address all points in detail later. First, we would like to ask for two clarifications.

You say that the paper shows that the traditional model ensemble (in the paper called ENS) does not improve the results for the two weeks of forecasts. We are not sure what you mean by this since the results shows that both the methods, traditional ensemble and neighbourhood, improve the results compared to the deterministic forecast (the control member).

Concerning the analysis of MVD and LWC, it is not clear to us how these parameters can be analysed, as we lack measurements. Could you please clarify this statement?

---

## Referee Comment (RC2) · Anonymous Referee #1 · 2 Nov 2017

Yes you are of course correct in that the ENS also improves the estimate. When I wrote that I was thinking of the Ensemble mean approach (EM), which did give slightly poorer results in Table 1 for production loss.

Yes about the MVD and LWC, the comment was less about a true "validation" of the parameters, but more a discussion of how they vary across the different ensembles. It is my understanding that the spread (Eq. 4) does not require any observed data, and therefore you could use it to discuss how much these terms vary say compared to their mean value. I think this is important as these terms are key to the icing model, so it is important to understand how much they vary in the different ensemble approaches you apply. Perhaps even just a simple standard deviation of the ensemble members could be used to investigate this.

---

## Referee Comment (RC3) · Anonymous Referee #2 · 7 Apr 2018

General comments 1. First of all I would like to state that it is an important subject with a clear application and obvious benefits. 2. How much new stuff is offered here? Neighbourhood method was suggested by Mittermaier [2014] but perhaps not applied. Using ensembles for short-term prediction of icing: To my knowledge this is new. 3. It is a weakness that only a very short time period studied: 2 sites in 2 weeks. One site has two icing events; the other one. So all in all the authors are presenting and discussing three (3) icing events and comparing those with an ensembles modelling. This is a somewhat weak evidence. 4. There is a short discussion at the end of paper. However, in order to judge the practical feasibility of using ensembles for ice prediction I miss more details on the calculations e.g. time elapsed for model runs and a discussion of whether it is feasible using the present approach or is likely to become feasible using

input from National weather services. By feasible I am thinking on feasible for owners of wind farms to e.g. obtain that service from specialized consultants.

Detailed comments 5. On page 8 l5 10 sites are mentioned, What happened to the 8 remaining? I miss a reason for eliminating these 8 sites 6. The neighbourhood principle is based on either following the landscape or using same height. I suppose you know where the sites are and might have added a short (anonymized) description of the landscape without revealing the location of the site? 7. Effect curve is termed power curve by many in the wind community. 8. It is not clearly stated whether de-icing equipment is included for the turbines studied.

Technical corrections 9. Figures are generally too small especially figures 4 to 10. 10. The choice of colour of the individual curves makes it hard to distinguish the curves especially figs 6, 9 and 11

---

## Author Response (AR1)

Dear reviewer,

We are grateful for your careful work and appreciate your detailed comments. Below you will find details how we will revise our manuscript based on your review.

With best regards,

Jennie Molinder (née Persson Söderman)

*General comments: This article looked at applying two well-known probabilistic techniques from the atmospheric science community to the problem of production loss of wind turbines due to icing. The methods were applied on the NWP model simulations that were used as input to the icing model, which in turn is used as input to the power production model. The paper shows that a traditional model ensemble does not improve results when tested over two wind farms for a two week period, but that using forecasts from neighboring points does improve the forecast.*

As discussed in the previous AC, both methods do improve the forecast skill. The referee also recognized this after our previous correspondence.

*The article is well structured and the topic is of significant interest to the Wind Energy community. However, the methods are not described in enough detail for reproduction, and then observational data that is used is not described sufficiently nor of sufficient length for it to provide a good validation of the methods being tested.*

Methods and observations will be described in more detail including
   - HarmonEPS setup and initialization – See answer to specific comment 1 & 2 below

- The icing model formulation including the effect of different water components types and ice loss - See answer to specific comment 5 & 6 below
- Table of observations with approximate location, mast or nacelle, observed parameter – Table 1 added to manuscript
- And more details according to your specific comments below.

We agree that the period is rather short, which is why we stressed the case study character of our work. The limitation resulted from the fact that global EPS data is required on the boundaries. These data are not operationally archived by ECMWF, only selected periods are available. We picked the most relevant period with available production observations. During this period, one of the three stations is containing two icing episodes and that, when combining all days and stations with icing, we have about 17 days of icing for validation. Given the scarcity of production observations and high computational costs of EPS runs, we still think that the results are robust for these meteorological conditions showing the possible benefit of using probabilistic forecasting for icing related production loss forecasts.

The extension of the dataset is not impossible, but not a simple task due to the need of global boundary data, high computational costs in running and storing the high-resolution ensemble and scarcity of icing and production observations. If regarded necessary, we are willing to find a solution.
We welcome guidance to this question by editor and reviewers.

*Finally, the paper mentions that the cloud parameters LWC and MVD are key inputs to the icing model, but they are not analyzed in terms of the spread of the probabilistic approaches, which seems like an important metric to analyze.*

A discussion about the mean and standard deviation for LWC and MVD will be added to the revised manuscript. Also addressing your specific comment below on the different water components, we will add a discussion on the impact of all water components and of the MVD and their spread to the results on the example of one icing event.

Added in the Section 3.2.2: Figure 11 with ENS mean MVD and mean spread for Site B. Since there was nearly no rain during the period, the MVD for the liquid water components follows the amount of LWC seen in figure 9c, and the spread was similar during the two weeks, only the figure showing MVD for the

solid water components was added.

Discussion about the MVD:

Added in section 3.2.1: During the icing event however, the ensemble members have a spread of around 25 % of the ENS mean value for liquid and solid water components, leading to a spread in the amount of built-up ice (Fig. 6c).

Added in Section 3.2.2: In Fig. 11a the ENS mean MVD for the solid water components can be seen for this site and the corresponding mean spread is shown in Fig. 11b. The liquid water components ENS mean MVD is not shown, as it follows the LWC amount in this case (Fig. 9c). Large values of MVD (Fig. 11a) coincide with large icing rates (Fig. 9a) stressing the role of MVD for the icing intensity. The large spread of the MVD around January 4 (Fig. 11b) can be connected to the ice load differences of the different ensemble members (Fig. 9a) and also to the simultaneous spread in the production loss forecast (Fig. 10b). This behaviour agrees with the effect of different MVD discussed in Davis et al. (2014).

Added in Section 3.2.2: This variation of the ensemble members is probably due to a large variation of the liquid and solid water content (between 25 and 50 % of the mean amount) resulting in a variation of the calculated MVD.

*Specific comments:*

*1. Section 2.1.1: How does the HarmonEPS used in this study differ from the operational usage of SMHI?*

1: The HarmonEPS currently used at SMHI are using a new model version cy40h1 and lagged ensemble members, but it should be noted that the operational version also includes physics perturbations. We will add a reference to the operational setup by Andrae et al. (2017, ALADIN-HIRLAM Newsletter No. 8, available from http://www.umr-

Added to Section 2.1.1: For the generation of the initial conditions for the ensemble members, the so-called PERTANA option is used, where the difference between the control analyses of HarmonEPS and ECMWF is added to the fields from the ECMWF EPS perturbed members. The HarmonEPS setup used here differs from the operational HarmonEPS currently running at SMHI in several aspects. The operational version uses a new model version (cy40h1.1), boundary conditions with the Scaled Lagged Average Forecast method, and also some physics perturbations (Andre and MetCoOp-Team, 2017).

*2. P4 L23-24: Do I read correctly that the models are all initialized using the ECMWF control member analysis, and only differ due to boundary conditions?*

2: We understand that our description on the initial conditions has to be clarified. In fact, we use the so-called PERTANA option, where the initial fields are combining the HarmonEPS control analysis with the fields from the ECMWF EPS perturbed members. We will revise our description.

Deleted in Section 2.1.1: and the perturbations for each ensemble member come from the boundary conditions of different members of the ECMWF EPS that are added to the analysis of the control member."

Added instead in Section 2.1.1: For the generation of the initial conditions for the ensemble members, the so-called PERTANA option is used, where the difference between the control analyses of HarmonEPS and ECMWF is added to the fields from the ECMWF EPS perturbed members.

*3. P5 L12: It would be interesting to see the impact of this averaging on the results. You state they are not here, but perhaps you could at least mention the improvement in error for the different variables in the text.*

3: The vertical interpolation was introduced for an earlier study and not validated again for this study. The earlier vertical interpolation used adiabatic lifting, except for inversion cases where a linear interpolation was

used. This caused jumps in wind speed etc. during regime changes. Thus, this more general vertical interpolation was introduced. Although undoubtedly interesting, we think that a more detailed discussion is beyond the scope of this article and plan to shorten this passage in order to keep the focus on the probabilistic forecasting.

For your interest, we add a plot with the wind on model level plus nacelle height, model level plus topographic correction plus nacelle height and our vertical correction. The RMSE is smallest for our vertical interpolation scheme.

[Figure]

**Figure 1. Wind speed at site A for the test period. Observations from nacelle are shown in black, modelled wind speed from model surface plus hub height in blue, modelled wind speed plus difference to real topography plus hub height in red, and the average of both modelled wind speeds in yellow.**

Deleted in Section 2.1.2:

Adiabatic lifting between model terrain and real terrain has yielded unsatisfying results.

Deleted in Section 2.1.2:

As important parameters for wind power, such as wind, temperature and moisture, vary strongly with the distance from the surface, a realistic choice for the vertical interpolation is vital. By choosing the averaging method from

Eq. 1 for height interpolation these error sources were reduced (not shown here).

Added in Section 2.1.2: This averaging method was used since

*4. P5 L12: What are all parameters, and which parameters was the lifting validated against?*

4: All model parameters used in the icing model are corrected. We will add a clarification. No validation for this method is included here. We think that it could be done in a separate study.

Added in Section 2.1.2: all atmospheric parameters that serve as input to the icing model.

*5. P6 L31-32: You note that different forms of water are fed separately through the model, how exactly is this carried out? Are you just running the model 4 times and summing the results? It is unclear to me how that would impact the accretion efficiency, which is a heat balance that depends on the mass flux of water impacting the structure.*

5: Yes, we are running the model several times and summing the results. For simplicity, we have treated all water components in the same way for alpha3. This assumption can be investigated. However, we think that it is not crucial for the main focus of the paper, namely the application of probablistic forecasting for wind power in cold climate. We will add a clarification on the calculation of alpha3 for the different water components.

Added in Section 2.1.3: Extended Eq. 2 to include the summing of the results and an ice loss term.

Added in Section 2.1.3: For simplicity, the accretion efficiency $\alpha_3$ is calculated in the same manner for the liquid and solid water components.

*6. P7 L2: Is the MVD only calculated for cloud water?*

6: The MVD is calculated for each of the water components since this is a necessary input to the icing model. In a revised version of the paper we will describe this more clearly.

Changed in Section 2.1.3: *Meteorological inputs needed for the ice calculations are temperature, wind speed, liquid water content, relative humidity and median volume droplet size. The latter is not directly available from the present NWP models, so a value is estimated using the liquid water content and the concentration of droplets.*

To in Section 2.1.3: Meteorological inputs needed for the ice calculations are temperature, wind speed, liquid/solid water content, relative humidity and median volume droplet size. The latter is not directly available from the present NWP models, so a value is estimated for all water components based on the liquid/solid water content and the concentration of droplets. The concentration of cloud droplets is set to a constant of 100 cm−1 except in the case of precipitation, when the number of droplets is instead based on output from the NWP model.

*7. P7 L4: What is the empirical ice shedding model?*

7: The ice shedding is basically a constant multiplied with the melting equation; this will be described better in a revised version of the paper.

Added in Section 2.1.3: The ice shedding is simulated by multiplying a constant of 8 with the melting term, increasing the process of removing the ice by a factor 8.

*8. P7 L15: In not familiar with the term effect curve, is this just the power curve?*

8: Yes, we are sorry for using the direct Swedish translation. This will be corrected.

Corrected

*9. P11 L26: You mention that there could be an issue with warm turbines, but you mentioned that you have a mixture of mast and nacelle data. Did you investigate if the bias was different between the two sources?*

9: The bias was not yet investigated in detail. We will examine the bias for

mast and nacelle data separately and add a remark to the manuscript.

Added in Section 3.1: This behavior might be caused by a spin-up problem in the model, since it is changing with forecast length. Additionally, a negative temperature bias that is not changing with forecast length can be attributed to the warm turbine affecting the measurements. A difference in the temperature bias between Mast and WT measurements amounts on average to -0.4 °C for mast and -1.1 °C for WT.

10. *P12 L10: could you list the % improvement of the wind speed and RH?*

10: Yes, the improvements for wind speed and RH is 7 and 12% respectively using ENSngb. This will be added in the article.

Deleted in Section 3.1: Similar behavior was found for the forecasted wind speed and relative humidity (not shown).

Added in Section 3.1: 7 % for the wind speed and 12 % for the relative humidity forecast error averaged for all sites and all forecasts

11. *P13 L9: The shift from 10 sites to 3 was confusing when reading this paper, perhaps remind the reader that you only had production data at 4 sites, and had to discard one. I am not sure why you only looked at two of the three sites here though.*

11: Thank you for the comment; we will describe this more clearly in a revised version of the paper.

Added in Section 2.2: Table 1. Observation sites with approximate latitude, description if the measurements were made from a mast or at the wind turbine nacelle (WT) and available production data (x).

Changed in Section 3.2: *Results from two of the three sites, here A and B, with consistent production observations will be presented in more detail here, and the average skill and spread for the different approaches for the*

*three sites are discussed.*

To in Section 3.2: Results from two of the three sites, here A and B, with consistent production observations will be presented in more detail in the following to point out some interesting features when using the probabilistic forecasts. Site C is not shown, but had icing during about three days of the two weeks. In Section 3.2.3, the average skill and spread for the different approaches for the three sites are discussed. Site A and C was during the two weeks affected by a frontal passage, while site A more experienced general cloudiness and was less affected by this front.

*12. Section 3.2: You discuss quite often the LWC values and how they impact the ice growth. It is unclear to me what this parameter is, since your model includes two types of liquid water hydrometeors and two solid hydrometeors. Could you describe what it is, and if it doesn't already make sure it includes all of the relevant hydrometeors.*

12: The LWC includes in this section refers to the liquid water content and not the solid parts. In our icing model, liquid water is a necessary condition for ice formation.

As you point out, it would be valuable to include all hydrometeors in the discussion. In a revised version of the manuscript we will, also according to your general comment above, add some discussion about the water components and related spread.

Changed: Figure 6c and 9c is changed from showing only the LWC to show both the total liquid water components and solid water components. In section 3.2.1 and 3.2.2 the different water components are discussed in more detail.

*13. Section 3.2: In the experiment period and available data section, you mention that you have production data from each turbine at each site. It is unclear how you aggregated the production data to get a singular value. Can you state approximately how many turbines were at each of the two sites you used?*

13. Yes, we will describe the aggregation of the production data and give an approximate number of wind turbines together with more detailed description of the sites in a revised manuscript.

Added in Section 3.2: From three of the sites, production data from each turbine is available for the period. The approximate location of the sites can be seen in Table 1 together with a specification if mast or nacelle (WT) observations were available and if the site had production data. The three sites with production data are at some distance from each other (Table 1). Site A is located on a hill with relatively high terrain west and north of the site, and somewhat lower terrain toward the south and east. In the location of site B there is instead similar terrain height to the south and west, while the terrain to the north and east is lower. The hill of site C extends mainly in the south-north direction, with lower terrain to the west and east. Site B and C have around 10 wind turbines, while site A consists of 20 turbines. All sites are surrounded with forest and with some lakes at lower levels. For each site one single value of production data is calculated by averaging production data from wind turbines without an error code. No de-icing system was used on the wind turbines included in the study.

Deleted in Section 3.2: For one of the four sites with production data, we found that wind speed measurements were unreliable, as the instrument was heavily affected by icing. This influence could be confirmed in webcam pictures. Thus, this stations wind speed data and its production data was omitted.

14. *While it is understandable that you cannot list the wind farms themselves, can you at least describe how far apart they are, and if they are exposed to similar weather patterns.*

14. We will add a characterisation and approximate location for each wind farm.

Added in Section 2.2: Table 1 with approximate location with a description

of the sites in the text

Added in Section 3.2: Sites A and C were affected by a frontal passage during the two weeks, while site A experienced general cloudiness and was less influenced by this front.

15. *Fig 6. It is hard to see the observations clearly, perhaps you could use a color like red that would stand out more.*

15: Thank you, we will make the figures clearer.

Changed: Red color for observations, grey color for CM instead of green and more relevant y-axis for the ice load figures.

16. *Figure 9: Why was there no ice during the beginning of the period, even though there was still a fairly large amount of LWC?*

16: Thank you for this observation! A quick analysis of the relevant meteorological parameters suggest this is because of very low wind speeds around the 29/12. We will examine this episode more carefully and add a comment about this in the revised manuscript.

Added in Section 3.2.2: In the beginning of the two weeks, there is one day with a large amount of liquid and solid water content, however, no ice started to build up during this day, mainly because of too low wind speeds.

17. *Table 1: What is the mean RMSE the mean of?*

17: It is the mean of the three sites. We will clarify this.

Removed in figure Table description: Mean

Added in table description: averaged over the three sites with production data.

**Anonymous Referee #2**

Dear reviewer,

We are grateful for your work and appreciate your comments. Below you will find details how we will revise our manuscript based on your review.

With best regards,

Jennie Molinder (née Persson Söderman)

*General comments 1. First of all I would like to state that it is an important subject with a clear application and obvious benefits. 2. How much new stuff is offered here? Neighbourhood method was suggested by Mittermaier [2014] but perhaps not applied. Using ensembles for short-term prediction of icing: To my knowledge this is new.*

*3. It is a weakness that only a very short time period studied: 2 sites in 2 weeks. One site has two icing events; the other one. So all in all the authors are presenting and discussing three (3) icing events and comparing those with an ensembles modelling. This is a somewhat weak evidence.*

3: We agree that the period is rather short, which is why we stressed the case study character of our work. The limitation resulted from the fact that global EPS data is required on the boundaries. These data are not operationally archived by ECMWF, only selected periods are available. We picked the most relevant period with available production observations. During this period, one of the three stations is containing two icing episodes and that, when combining all days and stations with icing, we have about 17 days of icing for validation. Given the scarcity of production observations and high computational costs of EPS runs, we still think that the results are robust for these meteorological conditions showing the possible benefit of using probabilistic forecasting for icing related production loss forecasts.

The extension of the dataset is not impossible, but not a simple task due to the need of global boundary data, high computational costs in running and storing the high-resolution ensemble and scarcity of icing and production observations. If regarded necessary, we are willing to find a solution. We welcome guidance to this question by editor and reviewers.

*4. There is a short discussion at the end of paper. However, in order to judge the practical feasibility of using ensembles for ice prediction I miss more details on the calculations e.g. time elapsed for model runs and a discussion of whether it is feasible using the present approach or is likely to become feasible using input from National weather services. By feasible I am thinking on feasible for owners of wind farms to e.g. obtain that service from specialized consultants.*

4: Thank you for the comment, we will include a discussion on feasibility and delivery times in the revised version. Since the introduction of the European INSPIRE directive, more and more national weather services are providing their forecasts as open data. Given the high computational costs of EPS, it seems beneficial, when national weather service can provide high-resolution EPS data for wind farm owners and specialized consults.

Changed in Section 4: *Finally, a NWP-ensemble forecast is computationally expensive to run and requires extensive infrastructure. Thus, often only national meteorological services can operationally produce ensemble forecasts. The open-data policy following the European INSPIRE directive will make these kinds of forecasts openly available. Then many users to include this forecast data into their own icing and production models which are far less computationally expensive. This will allow for a wide application of probabilistic forecasting for wind power in cold climates.*

To in Section 4: Finally, a NWP-ensemble forecast is computationally expensive to run and requires extensive infrastructure. Thus, often only national meteorological services can operationally produce ensemble forecasts. Many weather services currently run an operational ensemble forecast and the open-data policy following the European INSPIRE directive Many national weather ser- vices run currently an operational ensemble forecast that is often disseminated as open data following the European INSPIRE directive. The delivery time for such data are around 3 hours after analysis time, thus making it possible to use the 06-UTC model run for nextday wind power forecasts. Many users will then be able to include this forecast data into their own icing and production models, which are far less computationally expensive. This will allow for a wide application of probabilistic weather forecasting for wind power in cold climates.

*Detailed comments 5. On page 8 l5 10 sites are mentioned, What happened to the 8 remaining? I miss a reason for eliminating these 8 sites*

5: We have 10 sites with meteorological observations, but only three sites with production data. We apologize for the confusion in the description and will describe this in more detail in a revised version.

Added: Table 1 showing more clearly that three of the sites have production data.

Added in Section 2.2: From three of the sites, production data from each turbine is available for the period.

Changed in Section 3.2: *Results from two of the three sites, here A and B, with consistent production observations will be presented in more detail here, and the average skill and spread for the different approaches for the three sites are discussed.*

To in Section 3.2: Results from two of the three sites, here A and B, with consistent production observations will be presented in more detail in the following to point out some interesting features when using the probabilistic forecasts. Site C is not shown, but had icing during about three days of the two weeks. In Section 3.2.3, the average skill and spread for the different approaches for the three sites are discussed. Sites A and C were affected by a frontal passage during the two weeks, while site A experienced general cloudiness and was less influenced by this front.

*6. The neighbourhood principle is based on either following the landscape or using same height. I suppose you know where the sites are and might have added a short (anonymized) description of the landscape without revealing the location of the site?*

6: Yes, we will add a characterisation and approximate location for each wind farm.

Added to section 2.2: The approximate location of the sites can be seen in Table 1 together with a specification if mast or nacelle (WT) observations were available and if the site had production data. The three sites with production data are at some distance from each other (Table 1). Site A is located on a hill with relatively high terrain west and north of the site, and somewhat lower terrain toward the south and east. In the location of site B there is instead similar terrain height to the south and west, while the terrain to the north and east is lower. The hill of site C extends mainly in the south-north direction, with lower terrain to the west and east. Site B and C have around 10 wind turbines, while site A consists of 20 turbines. All sites are surrounded with forest and with some lakes at lower levels. For each site one single value of production data is calculated by averaging production data from wind turbines without an error code. No de-icing system was used on the wind turbines included in the study.

7. *Effect curve is termed power curve by many in the wind community.*

7: Ok thank you, we will change this.

Corrected

8. *It is not clearly stated whether de-icing equipment is included for the turbines studied.*

8: Thank you, it should be mentioned as you say that no de-icing system is used. We will add this to the manuscript.

Added to section 2.2: No de-icing system was used on the wind turbines included in the study.

*Technical corrections 9. Figures are generally too small especially figures 4 to 10. 10. The choice of colour of the individual curves makes it hard to distinguish the curves especially figs 6, 9 and 11*

9 & 10: Thank you, we will revise these aspects in the next version of the paper.

Changed: Now red color for observations, grey color for CM instead of green in figure 6,7,9 and 10. Red observation in figure 12 (old 11). More relevant y-axis for the ice load figures. Larger figures (we do not know how large the figures will be in a published version).

[revised manuscript text omitted]

---

## Author Response (AR2)

**Anonymous Referee #1, response 2.**

Dear reviewer,

Thank you for your suggested improvements and comments; we have revised the manuscript accordingly.

Best regards,

Jennie Molinder

1. *Consider moving the observed production and production loss descriptions to section 2.2 instead of section 2.2.1*

   Moved to section 2.2: The observed production loss was calculated from the ratio between the observed production and the potential production, given the observed wind speed and the ice-free power curves. A value of 30 % means 30 % less production than the potential production.

2. *pg. 6 Lines 9-11: I'm not sure this paragraph adds much as you describe bias correction being considered, but don't use it so it doesn't add much to the paper.*

   Removed from pg. 6: As a bias in the forecasted meteorological parameters could be damaging to the results, bias correction was considered at each time step before using the parameters as input to the icing and production model. However, no such correction is made, since no reliable information on bias for each station at each time step could be derived.

3. *Consider moving the two different neighborhood methods from section 2.1.2 to section 2.3. Also I am not quite sure I understand how eq. 1 changes with between the terrain following and non-following.*

   Removed from section 2.1.2: The neighbourhood method following Mittermaier (2014) is used in order to capture the local uncertainty of the NWP data. Averaging forecasts made at several grid points around an observation site results in a better forecast than one single forecast from kilometre-scale NWP. Furthermore, the spread of the forecast from the neighbouring grid points provides an estimate of the forecast uncertainty. Here, the 25 nearest grid points to an observation site are chosen as equally likely forecasts. Since these grid points are some kilometres apart from the turbine site, the height difference of the local topography can be several hundred meters. Two versions of the neighbourhood method was tested. In the first version, which is also the version presented in the result section (Section 3.2.3), the same height above sea level was used for all grid points, resulting occasionally in a height above ground much larger than the wind turbine height. The other version tested was a terrain-following method, where the same height above ground was used for all grid points. The different versions are further discussed in the result section.

   Added in section 2.3: The neighbourhood method following Mittermaier (2014) is used in order to capture the local uncertainty of the NWP data, e.g. the uncertainty in the representativeness of the forecast for a specific location of the wind turbines. Averaging forecasts 20 made at several grid points around an observation site results in a better forecast than one single forecast from kilometre-scale NWP. Furthermore, the spread of the forecast from the neighbouring grid points provides an estimate of the forecast uncertainty. Here, the 25 nearest grid points to an observation site are chosen as equally likely forecasts. Since these grid points are some kilometres apart from the turbine site, the height difference of the local topography can be several hundred meters. Two versions of the neighbourhood method was tested. In the first version, which is also the version presented in the result 25 section (Section 3.2.3), model data from the same height above sea level was used for all grid points, resulting occasionally in a height above ground much larger than the wind turbine height. The

other version tested was a terrain-following method, where model data from the same height above ground was used for all grid points, meaning that Δh in Eq. 1 is the same for all grid points. The different versions are further discussed in the result section.

Added for better understanding of the effect on Eq. 1: meaning that Δh in Eq. 1 is the same for all grid points

4. *Pg 7 lines 22-24: It would be nice to know the cylinder diameter that was used in your icing model, was it just the standard ISO diameter of 30 mm?*

From: based on an equation often referred to as the Makkonen equation: To: The model is following the ISO standard with a cylinder of 30 mm in diameter and is} based on an equation often referred to as the Makkonen equation:

5. *Pg 8 L 3: Was the interpolation just linear between 0 and 100%?*

Yes linear interpolation is used, this has been added in the paper.

6. *Pg 17 L10: It looks like you are missing a full citation at the end of the sentence.*

Thank you for observing this, it is now corrected.

[revised manuscript text omitted]